# Combine Harvester Bearing Fault-Diagnosis Method Based on SDAE-RCmvMSE

**DOI:** 10.3390/e24081139

**Published:** 2022-08-17

**Authors:** Guangyou Yang, Yuan Cheng, Chenbo Xi, Lang Liu, Xiong Gan

**Affiliations:** 1Institute of Agricultural Machinery, Hubei University of Technology, Wuhan 430068, China; 2Hubei Engineering Research Center for Intellectualization of Agricultural Equipment, Wuhan 430068, China

**Keywords:** entropy, bearing, vibration signal, SDAE-RCmvMSE, combine harvester

## Abstract

In the fault monitoring of rolling bearings, there is always loud noise, leading to poor signal stationariness. How to accurately and efficiently identify the fault type of rolling bearings is a challenge. Based on multivariate multiscale sample entropy (mvMSE), this paper introduces the refined composite mvMSE (RCmvMSE) into the fault extraction of the rolling bearing. A rolling bearing fault-diagnosis method based on stacked auto encoder and RCmvMSE (SDAE-RCmvMSE) is proposed. In the actual environment, the fault-diagnosis method use the multichannel vibration signals of the bearing as the input of stacked denoising autoencoders (SDAEs) to filter the noise of the vibration signals. The features of denoise signals are extracted by RCmvMSE and the rolling bearing operation-state diagnosis is completed with a support-vector machine (SVM) model. The results show that in the original test data, the accuracy rates of SDAE-RCmvMSE, RCmvMSE, and commonplace features of vibration signals combined with SVM (CFVS-SVM) methods are 99.5%, 100%, and 96% respectively. In the data with noise, the accuracy rates of RCmvMSE and CFVS-SVM are 97.75% and 93.08%, respectively, but the accuracy of SDAE-RCmvMSE is still 100%.

## 1. Introduction

Rolling bearings play an important role in agricultural machinery. Their vibration signals tend to be unsteady and nonlinear due to the harsh operating environment and complex bearing fault types [1,2]. Various nonlinear signal analysis and processing methods have been widely applied in bearing fault diagnosis [3]. Entropy has attracted the attention of researchers in more and more fields due to its unique advantages in the field of feature extraction, and a series of research achievements have been made [4,5]. Commonly used entropy methods include sample entropy [6], permutation entropy [7], dispersion entropy [8], etc. He [9] applied sample entropy and a support-vector machine (SVM) model to rolling bearing fault diagnosis to great effect. Cheng [10] used permutation entropy in bearing state monitoring, which effectively identified the working state of bearings. Rostaghi [11] used dispersion entropy for biological signal analysis, and the results showed that dispersion entropy algorithm had better stability and faster calculation speed and was less affected by mutation signals.

Dispersion entropy, sample entropy, and permutation entropy are all single-scale analysis methods based on time series, without considering nonlinear dynamics characteristics in multiple time scales. The feature acquisition ability of time series is limited and cannot reflect complex features. Azami [12] proposed refined multiscale dispersion entropy (RCMDE) on the basis of dispersion entropy and applied it to biological signal analysis. Comparison with multiscale dispersion entropy (MDE) showed that RCMDE has certain advantages over MDE in feature extraction and calculation speed. Costa [13] used multiscale sample entropy (MSE), which is a univariate method that can detect internal correlation. Multiscale sample entropy firstly uses multiscale reconstruction of time series and then calculates sample entropy at different reconstruction scales. This method has been used to measure the complexity of a single-channel physiological signal. In order to evaluate the structural complexity of multivariate signals using multiscale sample entropy, Ahmed [14] proposed multivariate multiscale sample entropy (MMSE) in multichannel simulation signals and biomedical signals. The MMSE algorithm considers the correlation within and across multiple data channels and evaluates them on multiple time scales, which provides an assessment of the dynamic richness of multichannel observations.

Deep learning is a signal detection and analysis method that has developed rapidly in recent years. The accuracy and generalization of deep learning in the field of fault diagnosis have been greatly improved [15,16,17,18]. Vinvent [19] introduced noise into a stacked autoencoder (SAE) [20] and formed a stacked denoising autoencoder (SDAE) algorithm. The algorithm extracted noiseless data from the data containing noise and extracted the deep features of the signal simultaneously. Li [21] used stacked autoencoders for rolling bearing fault diagnosis, and the results showed that the diagnosis was superior to BPNN and SVM models in terms of stability and accuracy.

In order to improve the stability and reliability of MMSE and give full play to the advantages of multivariate multiscale sample entropy in extracting nonlinear dynamic characteristics of multivariate signals, this paper proposes refined composite multivariate multiscale sample entropy (RCmvMSE). In order to improve the denoising performance of SDAEs, we propose an improved SDAE based on multiple noise distributions. The method fused the improved SDAE and RCmvMSE for rolling bearing fault diagnosis (SDAE-RCmvMSE). Firstly, the improved SDAE removed noise in the signals, then the RCmvMSE algorithm extracted the entropy characteristics of the signals. Finally, an SVM model was used to classify bearing faults.

## 2. Methods

### 2.1. Improved SDAE

DAE adds noise to the training data and extracts deep features from them, then reconstructs the original training data from the deep features. Get noisy data by adding noise to data sample, obtain deep features by encoder, and obtain the reconstruction result of the data sample by decoder. Finally, obtain the DAE model through minimizing the error between the data sample and reconstruction result. Stack all the DAEs to obtain the SDAE. In order to improve the generalization and denoising ability of the model, the improved SDAE uses Gaussian noises with different distribution centers to destroy the test data during model training [21], which is shown in Figure 1.

### 2.2. RCmvMSE

The main idea of refined composite technology is to generate a group of multivariate coarse-grained time series and obtain the average performance of these time series as the ultimate representation [22]. The refined composite multivariate multiscale sample algorithm includes the coarse-grained process and calculating multivariate sample entropy under various scale factors.

#### 2.2.1. Coarse-Grained Process

The traditional coarse-grained multiscale method does not fully consider the relationship between adjacent elements at both ends of each segment because of intercepting nonoverlapping segments. As the scale factor increases, the stability of the calculation results becomes worse [23], so adopt an improved refined composition technique of multivariate time series to solve this problem. The first step of the refined composite multivariate multiscale sample entropy algorithm is to generate a coarse-grained multivariate time series $z_{\alpha }^{\tau }$. The time series *Y* of the p channel is as follows:(1)$$\begin{array}{c} Y={\left\{y_{q,b}\right\}}_{b=1}^{c}\\ q=1,2,\ldots ,p \end{array}$$
where *C* is the length of each channel signal, and:$$z_{\alpha }^{\tau }=\{x_{\alpha ,q,i}^{\tau }\}$$
(2)$$\begin{array}{l} x_{\alpha ,q,i}^{\tau }=\frac{1}{\tau }\sum_{(i-1)\tau +\alpha }^{i\tau +\alpha -1}{\left(y_{q,b}-\frac{1}{\tau }\sum_{(i-1)\tau +\alpha }^{i\tau +\alpha -1}y_{q,b}\right)}^{2}\\ 1\leq i\leq \left\lfloor \frac{c}{\tau }\right\rfloor ,\alpha =1,2,\ldots ,\beta \end{array}$$
where τ is the delay coefficient and β the scale factor. For each τ, there is a corresponding coarse-grained sequence $Z_{\alpha }^{\tau }$. mvMSE only considers $Z_{1}^{\tau }$, so obtains relatively little information. The refined multivariate multiscale sample entropy algorithm uses coarse-grained sequences with different scales.

#### 2.2.2. Calculation for mvMSE

In order to calculate multivariate sample entropy [24] under a defined scale factor, the multivariate embedding vector is first generated. Suppose there is a p-channel signal $X={\left\{x_{q,i}\right\}}_{q=1,i=1}^{q=p,i=N}$, where N is the length of each coarse-grained time series. The multivariate embedded vector is defined as: (3)$$\begin{array}{cc} X_{m}(i)= & [x_{1,i},x_{1,i+\tau_{1}},\ldots ,x_{1,i+{(m_{1}-1)\tau }_{1}},\\ & x_{2,i},x_{2,i+\tau_{2}},\ldots ,x_{2,i+{(m_{2}-1)\tau }_{2}},\\ & \ldots ,x_{p,i},x_{p,i+\tau_{p}},\ldots ,x_{p,i+(m_{p}-1)\tau_{p}},] \end{array}$$
where M=[m1,m2,…,mp] is the embedding vector and the time-delay vector is $T=[\tau_{1},\tau_{1},\ldots ,\tau_{1}]$.

For the time series with *p* variables ${\left\{X_{q}\right\}}_{q=1}^{p}$, the calculation process of multivariate sample entropy is as follows.

(1)Calculate the multivariate embedding vector Xm(i)∈Rm, where i=1,2,…,N−n, n=max{m}×max{T}.(2)Calculate the distance between any two composite delay vectors Xm(i) and Xm(j) as the maximum norm.(3)For given Xm(i) and threshold *r*, calculate the quantity of *Pi* in d[Xm(i),Xm(j)]<r,i≠j.

The frequency of occurrence is
(4)$$\phi_{i}^{m}=\frac{1}{N-n}P_{i}$$
and define a variable as
(5)$$\phi^{m}(r)=\frac{1}{N-n}\sum_{i=1}^{N-n}\phi_{i}^{m}(r)$$

(4)Extend the dimension of the multivariate delay vector in (3) from *m* to *m* + 1 (dimensions of other variables remain unchanged).(5)Repeat steps (1)–(4) and calculate $\phi_{i}^{m+1}(r)$, then take the average $\phi^{m+1}(r)$ of all the *i* in *m* + 1 dimensions.(6)Finally, the multivariate sample entropy is calculated:(6)$$\mathrm{mvSE}(X,M,T,r)=-In\frac{\phi^{m+1}(r)}{\phi^{m}(r)}$$

Based on the improved recombination technique, each scale factor β has a corresponding coarse-grained sequence $Z_{\alpha }^{\beta }$. For each $Z_{\alpha }^{\beta }$, calculate $\phi_{\beta ,\alpha }^{m}$ and $\phi_{\beta ,\alpha }^{m+1}$ separately. Then, calculate ${\bar{\phi }}_{\beta ,\alpha }^{m}$ and ${\bar{\phi }}_{\beta ,\alpha }^{m+1}$ over the range α=1,2,…β. Finally calculate RCmvMSE: (7)$$\mathrm{RCmvMSE}(X,m,r,\tau ,\beta )=-In\frac{{\bar{\phi }}_{\beta ,\alpha }^{m}}{{\bar{\phi }}_{\beta ,\alpha }^{m+1}}$$

#### 2.2.3. Simulation Signal Analysis

In order to explain the characteristics of RCmvMSE in multivariate time series, RCmvMSE was applied to generated four-variable time series [25,26]: (1) all the four variable time series were Gaussian white noise; (2) three variables were Gaussian white noise and one variable was 1/*f* noise; (3) two variables were Gaussian white noise and two variables were 1/*f* noise; (4) all four variables are 1/*f* noise. The parameters of RCmvMSE are set as follows: embedding dimension *m* = 3, category *c* = 5, threshold *r* = 0.15, delay coefficient *d* = 1. The mean standard deviation of RCmvMSE of 20 scales for the four groups of time series with 3072 data length is shown in Figure 2.

It can be seen from Figure 2 that the RCmvMSE of the multivariate time series of each noise combination decreases with the increase of the scale factor. This shows that the complexity of time series under different scale factors is different. In the four-variable time series, with the decrease of the white noise time series variable, the variation range of RCmvMSE value decreases and the standard deviation increases correspondingly. This is because with the increase of 1/*f* noise, the dynamic characteristics of multivariate time series tend to be regular and concentrated, and the variability of multivariate time series is greater, leading to the increase in standard deviation. RCmvMSE can simultaneously obtain more information for each channel of multiple time series.

To compare the stability of RCmvMSE at different data lengths, 100 bivariate Gaussian white noise samples were used. The sample length of each group was 512, 1024, 1536, and 2048, respectively. The mean and standard deviation of RCmvMSE of 100 samples are shown in Figure 3. As can be seen from Figure 3a, RCmvMSE is more dispersed when the data length is low, and its value decreases with the increase in scale factor. When the data length is 2000, the value of RCmvMSE changes slightly. This shows that RCmvMSE is suitable for long time series. As shown in Figure 3b, the standard deviation of RCmvMSE decreases with the increase in data length and the decrease in scale factor. RCmvMSE is stable when the data length is 2048, so this article sets the sample length to 2048. 

## 3. Fault-Diagnosis Method Based on SDAE-RCmvMSE

### 3.1. SDAE-RCmvMSE Bearing Fault-Diagnosis Process

Based on the characteristics of SDAE and RCmvMSE and combined with the classification advantages of the support-vector machine (SVM) model [27,28,29], this paper proposes a bearing fault-diagnosis model based on SDAE-RCmvMSE, and its fault-diagnosis process is shown in Figure 4. The specific steps are as follows:(1)Collect original vibration signals through the data acquisition system installed on the bearing to be diagnosed.(2)Filter the noise in the vibration signal by improved SDAE.(3)RCmvMSE is used to extract the entropy features of vibration signals to obtain training set and testing set.(4)Sending the training set to SVM, parameters of SVM are updated by the error between expected outputs and actual output. This step is iterated repeatedly until the classifier accuracy requirement or the maximum number of iterations is reached. Then, the SVM model is completed and the optimal parameters are obtained.(5)The SVM model completed by training is used to classify the samples in the feature testing set, so as to obtain the state types of bearings to be diagnosed.

**Figure 4 entropy-24-01139-f004:**
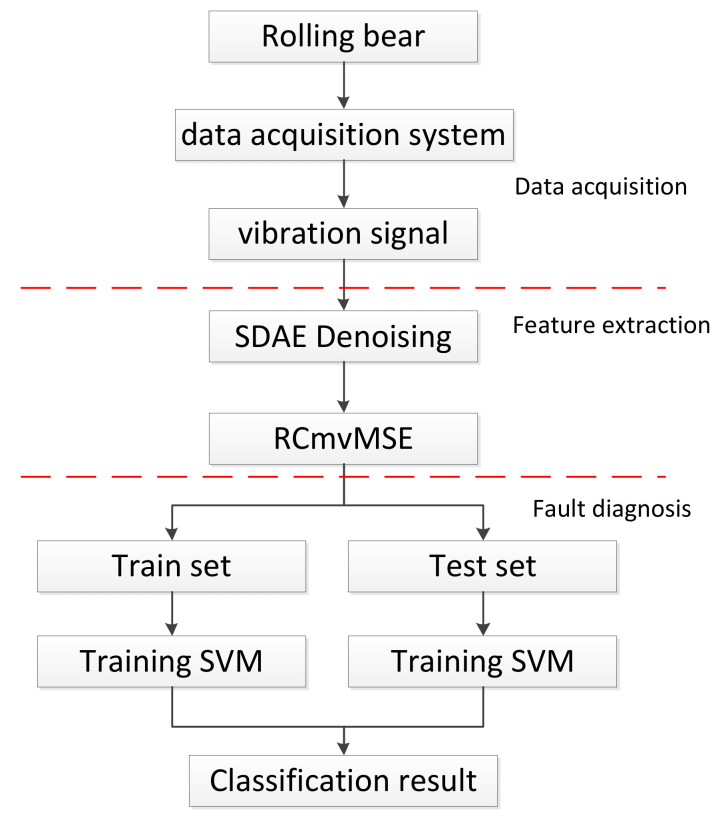
Flowchart of bearing fault diagnosis.

### 3.2. Bearing Fault-Diagnosis Experiment

To verify the effectiveness of the algorithm, it is applied to the single fault-state identification experiment of a combine harvester detached from a roller bearing. The test equipment in this paper is a threshing drum assembly of a combine harvester. The test platform is mainly composed of a motor driving part, threshing assembly, and data acquisition system. In the test, data acquisition was completed by TM0782A acceleration sensor, NJK-5002C speed sensor, USB-6002 data acquisition card, computer, and Labview 2019 software. Figure 5a,b show the positions of the acceleration sensor and speed sensor. The voltage sensitivity of the #1 to #4 accelerometers is 101.6 mV/g, 99.1 mV/g, 101.2 mV/g, and 101.3 mV/g, respectively. 

The test bearing is a 6307 deep-groove ball bearing at the rear end of the threshing drum. The outer ring of the bearing is fixed on the bearing seat, and the inner ring rotates with the threshing drum shaft. Other bearings in the system are normal bearings. Three common fault types of 6307 deep-groove ball bearings were considered in this test: inner ring crack (IRF), outer ring crack (ORF), and compound crack (CF). The detailed parameters of each fault are shown in Table 1.

Bearing faults of different types and sizes were simulated by edM. Figure 6 shows the rolling bearings with inner, outer, and inner ring composite faults, respectively. The vibration signals of bearing state were collected by data acquisition system under four working conditions, namely, the speed of threshing drum was stable at 80 r/min, 160 r/min, 240 r/min, and 320 r/min, with sampling frequency of 5120 Hz and sampling time of each group of states was 50 s. The time-domain waveforms of some fault types are shown in Figure 7.

### 3.3. Analysis of RCmvMSE

The collected data are sequentially divided into data samples with a length of 2048, and there are 300 data samples in each fault state and 3000 data samples in normal state. RCmvMSE parameters are set as: embedding dimension *m* = 3, category *c* =5, threshold *r* = 0.15, delay coefficient *d* = 1.

It can be seen from Figure 8 that RCmvMSE decreases with the increase in the scale factor. However, when the scale factor is greater than 11, RCmvMSE tends to be stable and the difference is small. When the scale factor is less than 9, the value separation of RCmvMSE is also obvious [30,31].

Five samples of IRF07 were taken, a sample window with a data length of 2048 and a step size of 32 was taken to sweep the sample, and the RCmvMSE curve with a scale factor of 15 was drawn, as shown in Figure 9.

It can be seen from Figure 9 that the periodicity of RCmvMSE is very significant.

In order to research the impact of scale factors on model accuracy, 30 samples from each type were used as test samples. RCmvMSE was used to extract the entropy characteristics of the test samples under different scale factors, and then the trained SVM was used for classification. The results are shown in Figure 10. It can be seen that the accuracy of the model increases with the increase in the scale factor. When the scale factor is greater than 15, the accuracy of the model tends to be stable, but when the scale factor is greater than 20, the accuracy of the model decreases slightly. The reason for the decrease in accuracy may be that the scale factor is too large to produce information redundancy. Therefore, the scale factor of the RCmvMSE is set to 15 in this paper.

### 3.4. Analysis of Results

#### 3.4.1. Analysis with Origin Data

SVM classifier was used to classify features extracted by RCmvMSE to realize fault classification of rolling bearings. Firstly, 280 samples for each fault type of data and 2000 normal data samples were taken as the training set. The remaining samples were taken as the testing set. Meanwhile, normal, IRF07, IRF10, …, CF05, and CF10 are identified as 1, 2, …, and 11, respectively. Then, the training samples were input into the SDAE-RCmvMSE model based on SVM for training.

In order to solve the impact of unbalanced samples on the model, adjust the weight of fault-state data samples during model training, so that the error loss caused by fault samples will have a greater impact on model parameters. Because of the SDAE-RCmvMSE model, the vibration signal data has a certain denoising ability. Therefore, the trained SDAE-RCmvMSE model was tested with noised vibration signal data and different degrees of noised vibration signal data. At the same time, for further verification of the advantages of SDAE-RCmvMSE, the vibration data in the third channel of the same training sample were used to train SVM based RCmvMSE, and the tests were also carried out in noise-free and noise-added data. To better display the performance of SDAE-RCmvMSE, commonplace features of vibration signals combined with SVM (CFVS-SVM) were used for bearing fault diagnosis in this paper. Common features include: time domain, frequency domain, and time frequency domain. The time-domain features in this paper are 18, such as mean, standard deviation, peak-to-peak value, median, quartile difference, waveform factor, peak factor, impulse value, and margin. The frequency domain features include the amplitude of Fourier transform, median of mean, quartile difference, and percentile difference, etc. The time-frequency domain features are six energy features and their normalized energy features obtained by five-order wavelet transform. Results of RCmvMSE, CFVS-SVM, and SDAE-RCmvMSE models in the noise-free data test samples are shown in Table 2. RCmvMSE reached 100% accuracy in noiseless data, which is much higher than CFVS-SVM. The reasons may be as follows: 1. the test environment is ideal, and the feature of each type of fault is obvious; 2. the fault type is simple and there is no compound fault. SDAE-RCmvMSE and CFVS-SVM have the lowest diagnostic accuracy of IRF10 and IRF07, which are both 85%. Meanwhile, the test time of single samples of RCmvMSE and SDAE-RCMVMSE was 0.98 s and 1.13 s, respectively.

#### 3.4.2. Analysis with Noise Data

Using Gaussian noise with a mean of 0 and a variance of 0.1 to randomly destroy the original sample with a probability of 0.10, a noise test sample was obtained. Noise test samples were used to test RCmvMSE, CFVS-SVM, and SDAE-RCmvMSE models, and the results are shown in Table 3. As can be seen from Table 3, the accuracy of RCmvMSE and CFVS-SVM is 97.75% and 93.08%, respectively. Under the testing conditions of noisy data samples, the accuracy is greatly reduced. The accuracy of IRF12 and ORF07 reaches 100%, and its F1 measurement is 0.9137 [32], indicating that RCmvMSE has a general performance in the data samples containing noise. Compared with the above two algorithms, the accuracy of SDAE-RCMVMSE reaches 100% and the F1 measurement is 1, indicating that the fault-diagnosis ability of the SDAE-RCMVMSE model is significantly better than the CFVS-SVM and RCmvMSE models in the signal containing noise. For inner ring failure, that of the original signal is shown in Figure 11, after adding noise the time-domain waveform of the original signal is shown in Figure 12, and that of the original signal after denoising is shown in Figure 13. It can be seen from Figure 11, Figure 12 and Figure 13 that the SDAE model has excellent denoising performance. The reason for the accuracy of 100% may be that the difference of each fault type is large and the test conditions are relatively ideal.

In terms of information-processing time, the single-sample test time of RCmvMSE was about 0.94 s and the single-sample test time of SDAE-RCMVMSE was about 1.13 s. The single-sample test time of RCmvMSE was significantly lower than that of SDAE-RCMVMSE. SDAE-RCMVMSE has excellent performance in noisy data, but the test time is relatively long. The test time can be reduced by reducing the scale factor, but its stability may be affected to some extent, and further research is needed.

#### 3.4.3. Visualization of Classification Results

For all test samples in the rolling bearing data set, the visual distribution results of features extracted by SDAE-RCMVMSE after t-SNE processing [33] are shown in Figure 14. It can be seen from the figure that this method has great feature distribution and the samples between classes are relatively dispersed. Except for the compound fault 2 category, the samples within other classes have a high degree of aggregation. In compound fault 2, the intraclass aggregation effect is poor, but it does not affect the accuracy of the diagnosis results.

#### 3.4.4. Model Performance Analysis under Different SNR

In order to analyze the relationship between Gaussian noise with a mean of 0 and a variance of 0.1 in the original data and the diagnostic accuracy of the model, white Gaussian noise was added to the original vibration data according to the ratio of 0%, 10%, 20%, and 30% (the mean of 0 and variance of 0.1), and the average diagnostic accuracy of different fault diagnosis methods calculated within 50 running times. As shown in Figure 15, the accuracy of SDAE-RCMVMSE, RCmvMSE, and CFVS-SVM all declined with the increase in the proportion of Gaussian noise. When the proportion of Gaussian noise is less than or equal to 0.30, the accuracy of SDAE-RCMVMSE algorithm is always stable at 99.7–98.8%, while the accuracy of the RCmvMSE algorithm decreases slightly and is greater than 93%. This shows that the SDAE-RCmvMSE model has better fault-diagnosis ability than CFVS-SVM and RCmvMSE in the signal containing noise, and has strong antinoise performance.

## 4. Conclusions

In this paper, a bearing fault-diagnosis method based on SDAE-RCMVMSE is proposed and verified by a bearing fault simulation test on a combine harvester test platform. The experimental results verify the effectiveness of the method. The main conclusions are as follows:(1)The SDAE model can effectively remove noise points in the signal, laying a foundation for accurate extraction of RCmvMSE;(2)The improved SDAE model introduces Gaussian noises with different distribution centers, which greatly improves the denoising ability of the model and makes the model more robust;(3)RCmvMSE feature extraction considers the relevant information of each channel in multivariate variables, and the extracted entropy can better reflect the changes of multivariate signals and has great stability.

The SDAE-RCmvMSE fault-diagnosis method proposed in this paper has excellent antinoise performance, high recognition rate, and strong robustness, which provides a new technical means for fault monitoring and diagnosis of rotating mechanical components. Subsequent studies will further improve the real-time performance of the algorithm and the effectiveness of complex fault diagnosis.

## Figures and Tables

**Figure 1 entropy-24-01139-f001:**
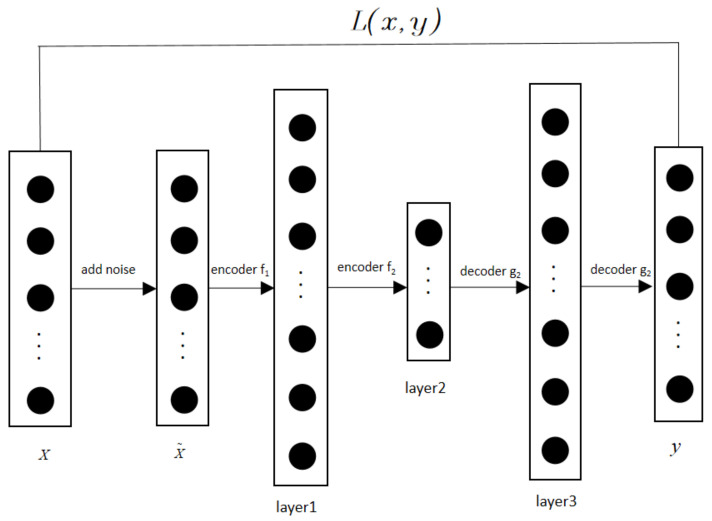
Structure of improved SDAE model. Note: x is the data sample, $\tilde{X}$ is the noise data, layer 1 and layer 2 are the feature representation layers, layer3 is the feature reconstructed presentation layer, and y is the output.

**Figure 2 entropy-24-01139-f002:**
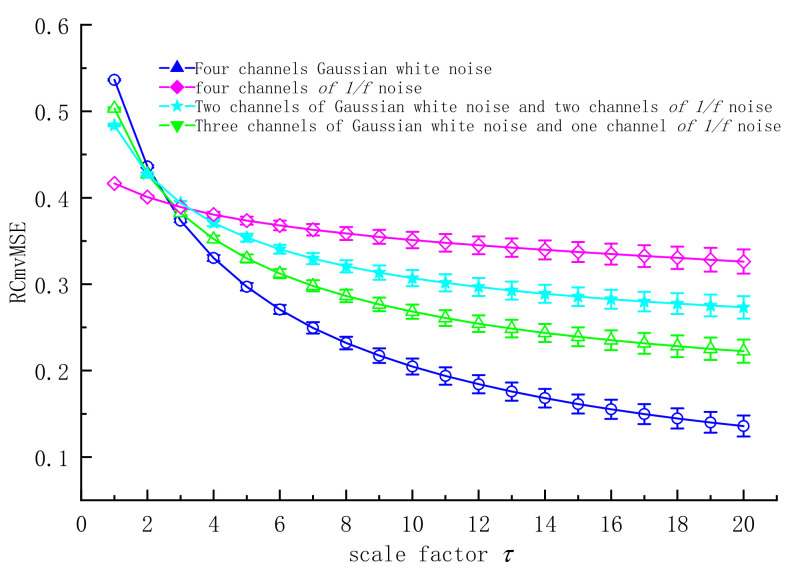
Mean chart of RCmvMSE.

**Figure 3 entropy-24-01139-f003:**
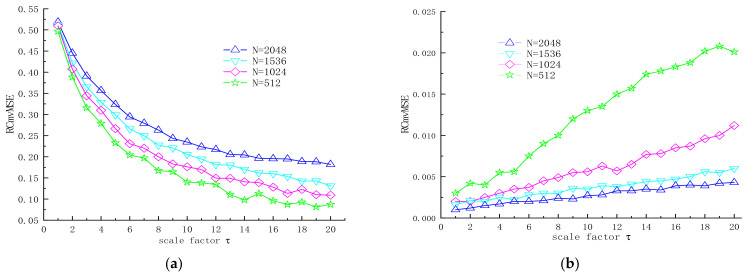
Mean and standard deviation of RCmvMSE in different signal lengths. (**a**) Mean value of RCmvMSE; (**b**) standard deviation of RCmvMSE.

**Figure 5 entropy-24-01139-f005:**
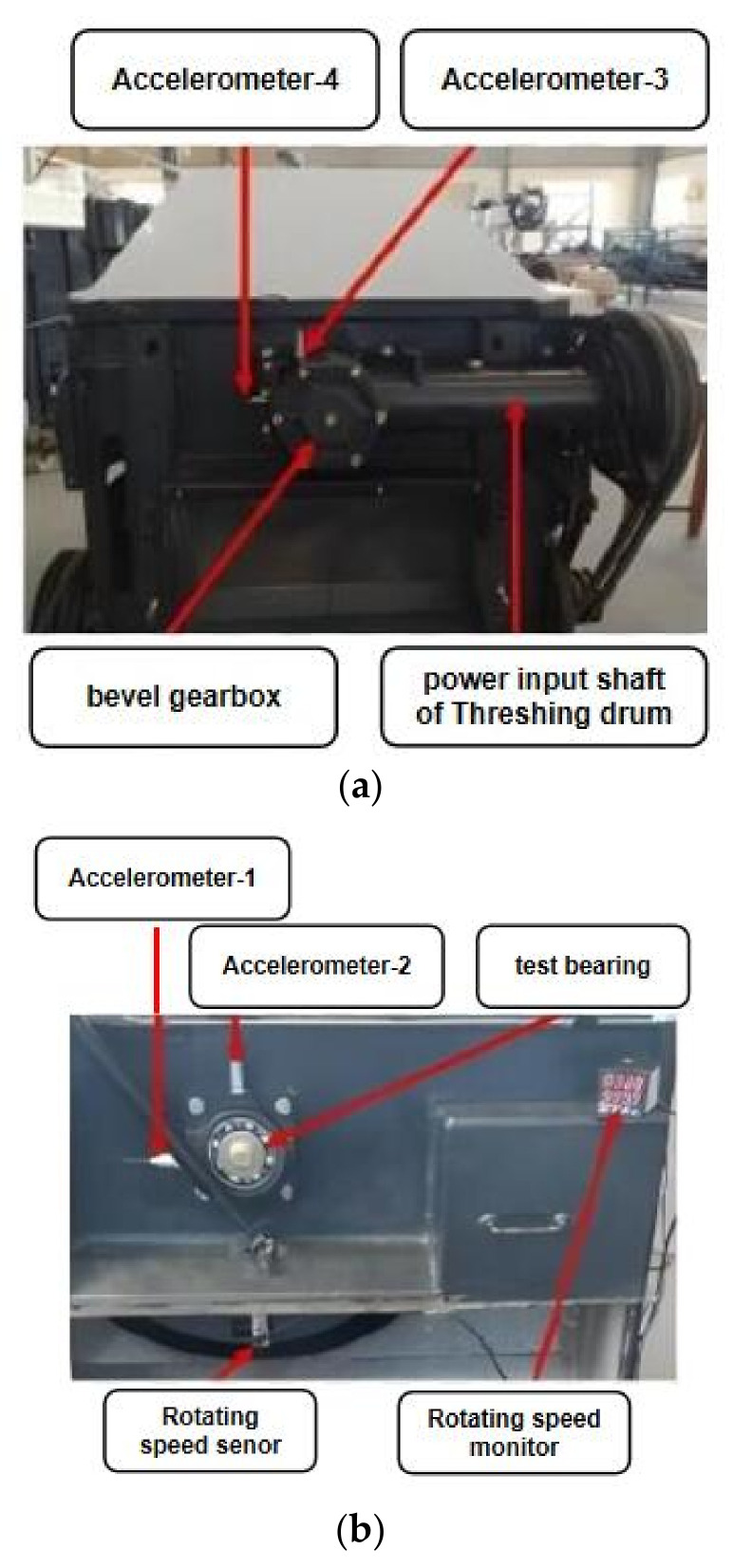
Schematic diagram of sensor installation of combine harvester test platform. (**a**) Installation diagram of acceleration sensor of bearing under test. (**b**) Installation diagram of acceleration sensor of bevel gearbox.

**Figure 6 entropy-24-01139-f006:**
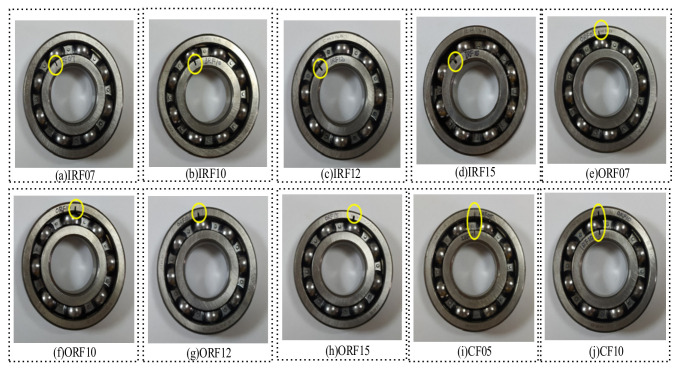
Different types of faulty bearings.

**Figure 7 entropy-24-01139-f007:**
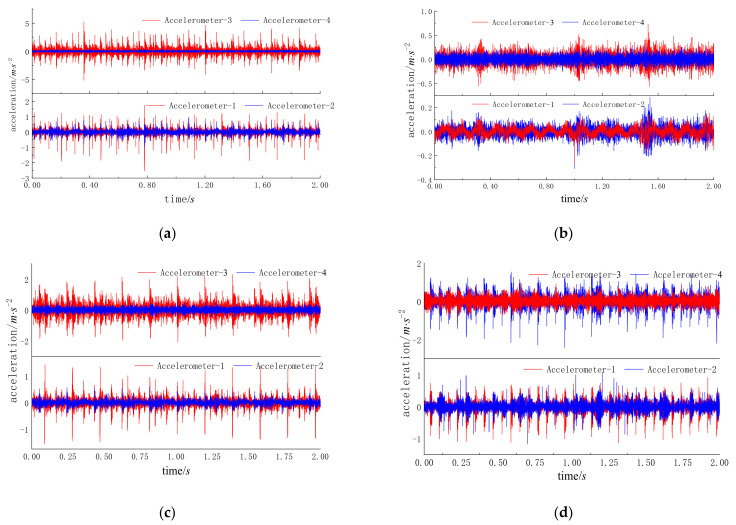
Time domain waveforms of different faults. (**a**) CF05; (**b**) Normal; (**c**) IRF15; (**d**) ORF15.

**Figure 8 entropy-24-01139-f008:**
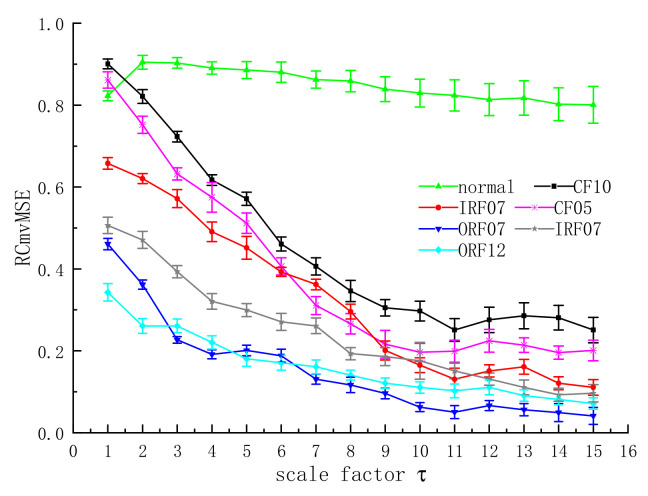
Mean standard deviation chart of RCmvMSE.

**Figure 9 entropy-24-01139-f009:**
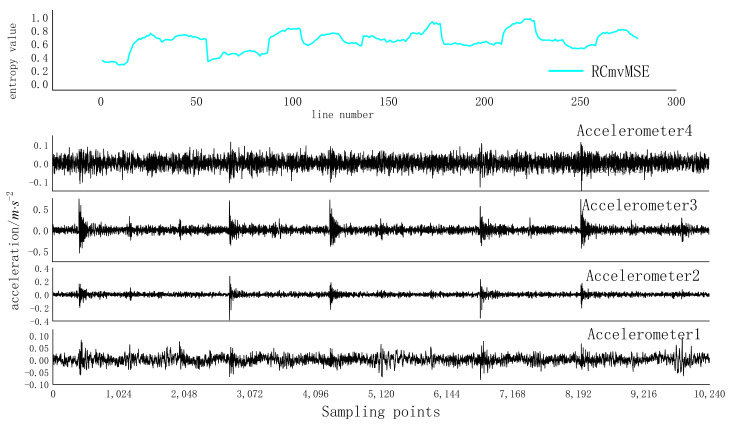
RCmvMSE curves under inner ring failure 1 state.

**Figure 10 entropy-24-01139-f010:**
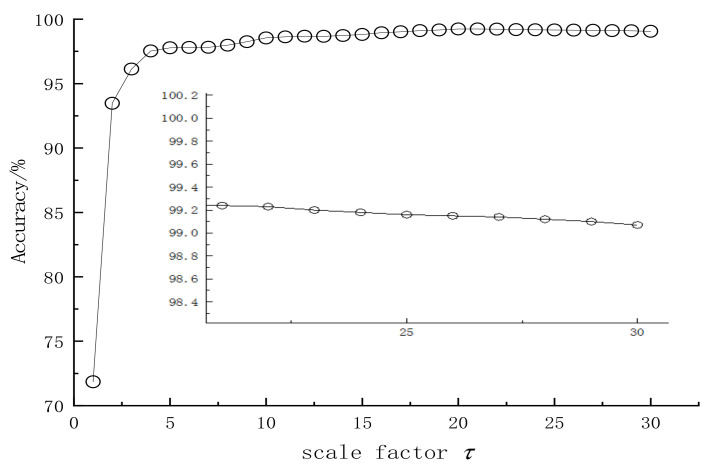
Model accuracy varies with the scale factor.

**Figure 11 entropy-24-01139-f011:**
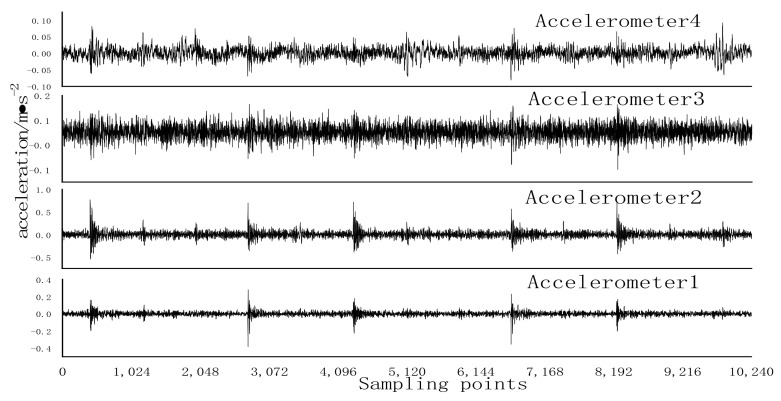
Original signal time domain waveform.

**Figure 12 entropy-24-01139-f012:**
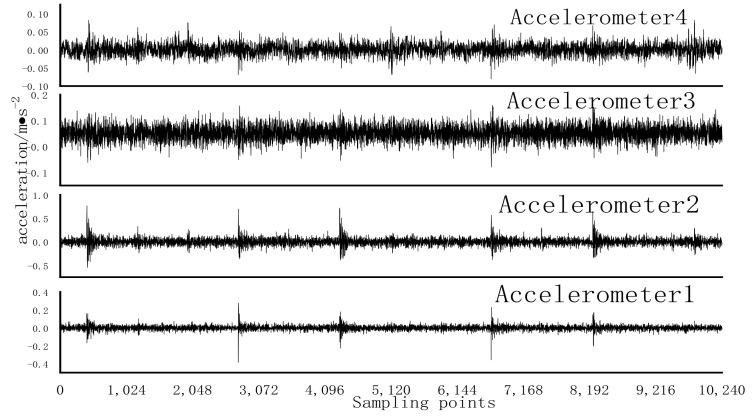
Time domain waveform after the addition of noise to the original signal.

**Figure 13 entropy-24-01139-f013:**
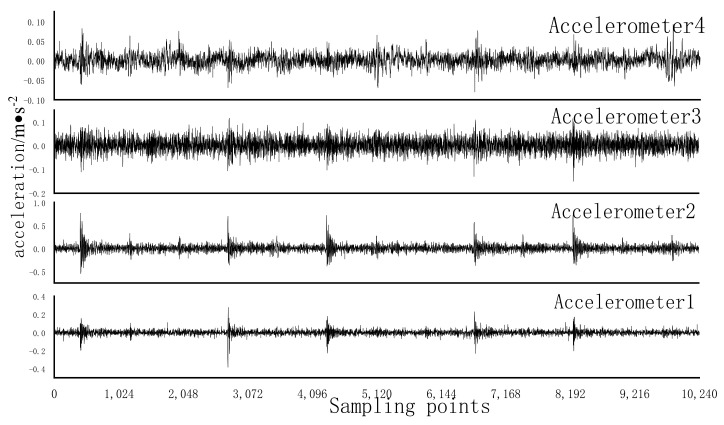
Time domain waveform after denoising.

**Figure 14 entropy-24-01139-f014:**
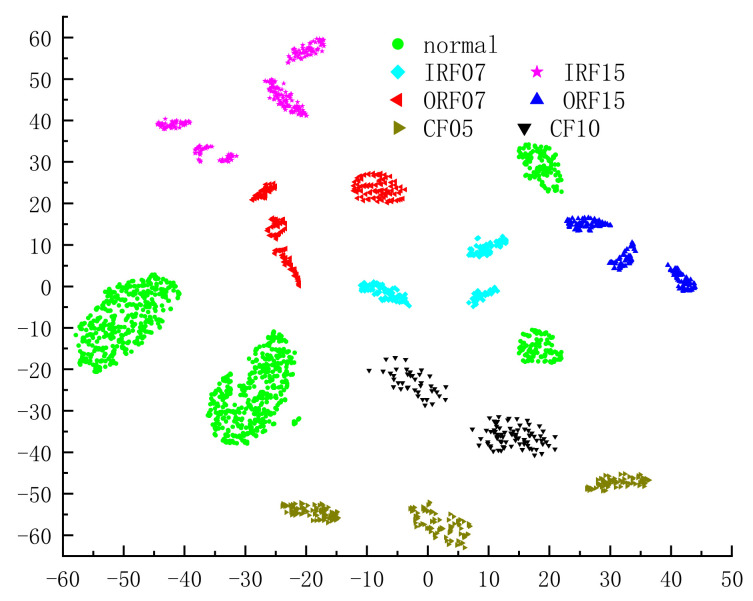
Visualization results of feature distribution.

**Figure 15 entropy-24-01139-f015:**
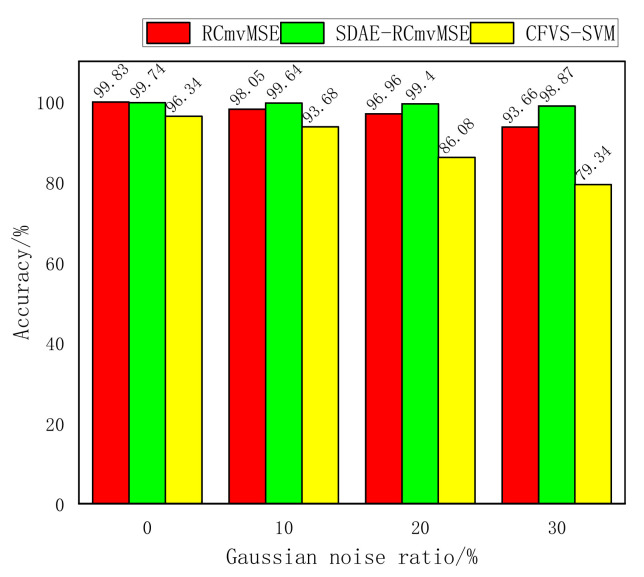
The relationship between the accuracy and the proportion of Gaussian noise.

**Table 1 entropy-24-01139-t001:** Parameters of different states.

State	Fault Width/mm	Fault Depth/mm
IRF07	0.7	3.7
IRF10	1.0	3.7
IRF12	1.2	3.7
IRF15	1.5	3.7
ORF07	0.7	3.2
ORF10	1.0	3.2
ORF12	1.2	3.2
ORF15	1.5	3.2
CF05	0.5/0.5	3.2/3.7
CF10	1.0/1.0	3.2/3.7

**Table 2 entropy-24-01139-t002:** Diagnostic results of SDAE-RCmvMSE, RCmvMSE, and CFVS-SVM in noise data.

State	Number of Samples	Prediction Accuray/%		
		RCmvMSE	SDAE-RCmvMSE	CFVS-SVM
Normal	1000	100.00	100.00	96.70
IRF07	20	100.00	90.00	85.00
IRF10	20	100.00	85.00	95.00
IRF12	20	100.00	100.00	100.00
IRF15	20	100.00	100.00	85.00
ORF07	20	100.00	100.00	95.00
ORF10	20	100.00	100.00	100.00
ORF12	20	100.00	100.00	100.00
ORF15	20	100.00	100.00	90.00
CF05	20	100.00	95.00	90.00
CF10	20	100.00	100.00	85.00
Total	1200	100.00	99.50	96.00

**Table 3 entropy-24-01139-t003:** The diagnosis results of SDAE-RCmvMSE, RCmvMSE, and CFVS-SVM under noise data.

State	Number of Samples	Prediction Accuracy/%		
		RCmvMSE	SDAE-RCmvMSE	CFVS-SVM
Normal	1000	99.10	100.00	94.20
IRF07	20	95.00	100.00	85.00
IRF10	20	95.00	100.00	90.00
IRF12	20	100.00	100.00	90.00
IRF15	20	80.00	100.00	80.00
ORF07	20	100.00	100.00	90.00
ORF10	20	90.00	100.00	85.00
ORF12	20	90.00	100.00	90.00
ORF15	20	85.00	100.00	90.00
CF05	20	85.00	100.00	85.00
CF10	20	90.00	100.00	90.00
Total	1200	97.75	100.00	93.08

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
