# Peer review of "Combine Harvester Bearing Fault-Diagnosis Method Based on SDAE-RCmvMSE"

_entropy, 2022, doi:10.3390/e24081139_

Round 1

Reviewer 1 Report

Title: Combined harvester bearing fault diagnosis method based on SDAE-RCmvMSE

In this paper, the authors studied the refined composite multivariate multiscale sample entropy, where a rolling bearing fault diagnosis method based on stacked auto encoder is proposed.  Several theoretical results are obtained. The paper is well structured and the main results seem sound. 

1. The presentation of this paper is clouded by mathematical and notational ambiguities and complexities which render the contribution of the paper difficult to grasp with accuracy.

2. More detailed comparisons with some existing methods or references should be made in order to highlight the main contributions of this paper. 

3. Some figures and tables should be well presented and organized for a clear presentation. 

4. Some mathematical symbols or derivations should be carefully checked and revised, and some symbols should be defined beforehand. 

5. Moreover, some typos are found in the reference parts, see [12], the format of the references should be unified.

Author Response

DearReviewers:

Thank you for your letter and for the reviewers’ comments concerning our manuscript entitled “ Combined harvester bearing fault diagnosis method based on SDAE-RCmvMSE” (entropy-1824540).Those comments are all valuable and very helpful for revising and improving our paper, as well as the important guiding significance to our researches. We have studied comments carefully and have made correction which we hope meet with approval.

Response to the questions one by one:

  1. The presentation of this paper is clouded by mathematical and notational ambiguities and complexities which render the contribution of the paper difficult to grasp with accuracy.

The mathematical formulations of this article have been adjusted to account for the individual variables. This paper mainly improves the traditional method, adds Gaussian noise of different distribution in sdae, and proposes RCmvMSE.

  1. More detailed comparisons with some existing methods or references should be made in order to highlight the main contributions of this paper. 

Compared with RCmvMSE and CFVS-SVM, sdae-RCmvMse in this paper has higher accuracy under noisy and noise-free data.

  1. Some figures and tables should be well presented and organized for a clear presentation. 

The chart has been redrawn and the text in the chart has been adjusted

  1. Some mathematical symbols or derivations should be carefully checked and revised, and some symbols should be defined beforehand. 

Mathematical formulas have been revised and relevant variables have been defined.

  1. Moreover, some typos are found in the reference parts, see [12], the format of the references should be unified.

The format of the references has been revised

Reviewer 2 Report

Bearing fault diagnosis based on SDAE-RCMVMSE is proposed. It is  verified through bearing fault simulations on a combined harvester test platform. The experimental results are agreeable. The main conclusions are : SDAE model can removes noise, It introduces gaussian noise with different distribution centers, RCmvMSE feature extraction considers the relevant information of each channel multivariate variables.The extracted entropy reflects has great stability. The SDAE-RCmvMSE has good noise performance and strong robustness.

Author Response

Dear Reviewers:

Thank you for your letter and for the reviewers’ comments concerning our manuscript entitled “ Combined harvester bearing fault diagnosis method based on SDAE-RCmvMSE” (entropy-1824540).Those comments are all valuable and very helpful for revising and improving our paper, as well as the important guiding significance to our researches. We have studied comments carefully and have made correction which we hope meet with approval. The English language and style of this article has been revised.

Reviewer 3 Report

This paper proposed a rolling bearing fault diagnosis method based on stacked auto encoder and refined composite multivariate multiscale sample entropy. It is of interest. However, the authors should address the following points to further improve the quality.

1. The language of this manuscript needs further condensing and improvement. For example, “train set” should be revised as “training set”.

2. The advantage of the proposed method should be clearly summarized, and the contributions of this study should be highlighted in the paper.

3. What is the motivation for designing such a hybrid learning model based on SAE and RCmvMSE for fault detection of rolling bearing?

4. In abstract, “Stacked Auto Encoder” should be revised as “Stack Denoise Auto Encoder”. There are a lot of typos like this. Please proofread carefully.

5. Literature review on the approaches of fault diagnosis is limited. More recently-published papers in this field should be discussed. The authors may be benefited by reviewing more papers such as 10.1016/j.ymssp.2022.109569 and 10.1109/TIM.2022.3181894.

6. The quality of some figures should be improved, Such as the text in Fig.1, Fig.2 and Fig.5.

7. The specific structure and hyper-parameter settings of each component in the model are not clear, please list them.

Author Response

Dear Reviewers:

Thank you for your letter and for the reviewers’ comments concerning our manuscript entitled “ Combined harvester bearing fault diagnosis method based on SDAE-RCmvMSE” (entropy-1824540).Those comments are all valuable and very helpful for revising and improving our paper, as well as the important guiding significance to our researches. We have studied comments carefully and have made correction which we hope meet with approval.

Response to the questions one by one:

  1. The language of this manuscript needs further condensing and improvement. For example, “train set” should be revised as “training set”.

The English language and style of the paper has been heavily revised.

  1. The advantage of the proposed method should be clearly summarized, and the contributions of this study should be highlighted in the paper.

In order to improve the stability and reliability of MMSE and give full play to the advantages of multivariable Multiscale Sample Entropy in extracting nonlinear dynamic characteristics of multivariable signals.This paper proposes Refined Composite multivariable Multiscale Sample Entropy (RCmvMSE). In order to improve the denoising performance of SDAE, this paper proposes an improved SDAE based on multiple noise distributions.

  1. What is the motivation for designing such a hybrid learning model based on SAE and RCmvMSE for fault detection of rolling bearing?

The improved sdae can better remove noise in data, and RCmvMSE can better extract features

  1. In abstract, “Stacked Auto Encoder” should be revised as “Stack Denoise Auto Encoder”. There are a lot of typos like this. Please proofread carefully.

The English language and style of the paper has been heavily revised.

  1. Literature review on the approaches of fault diagnosis is limited. More recently-published papers in this field should be discussed. The authors may be benefited by reviewing more papers such as 10.1016/j.ymssp.2022.109569 and 10.1109/TIM.2022.3181894.

I watched some latest articles about fault diagnosis,《Self-Adaptation Graph Attention Network via Meta-Learning for Machinery Fault Diagnosis with Few Labeled Data》and《Discriminative feature learning using a multiscale convolutional capsule network from attitude data for fault diagnosis of industrial robots[J]. Mechanical Systems and Signal Processing》benefited me a lot.

  1. The quality of some figures should be improved, Such as the text in Fig.1, Fig.2 and Fig.5.

The pictures in the paper have undergone many modifications

  1. The specific structure and hyper-parameter settings of each component in the model are not clear, please list them.

The parameters of RCmvMSE are set as follows: embedding dimension m=3, category c=5, threshold r=0.15, delay coefficient d=1.sample length=2048.

Round 2

Reviewer 1 Report

No further comments. 

Reviewer 3 Report

The revision has addressed all my issues. The quality has improved a lot after revision. I recommend it for publication.